# Hilbert-Coding Metasurface for Diverse Electromagnetic Controls

**DOI:** 10.3390/ma15175913

**Published:** 2022-08-26

**Authors:** Jianjiao Hao, Fuju Ye, Ying Ruan, Lei Chen, Haoyang Cui

**Affiliations:** College of Electronics and Information Engineering, Shanghai University of Electric Power, No. 2588 Changyang Road, Yangpu District, Shanghai 200090, China

**Keywords:** metamaterial, metasurface, passive metasurface, coding metasurface, Hilbert-curve

## Abstract

Metamaterials, or metasurfaces, allow the flexible and efficient manipulation of electromagnetic (EM) wave. Although the passive coding metasurfaces have achieved a great deal of functionality, they also need a complex design process. In this paper, we propose Hilbert-coding metasurfaces for flexible and convenient EM regulation by arranging Hilbert-coding metamaterial units of different orders. To demonstrate this behavior, we designed 12 metasurfaces, then fabricated and measured 6 samples. Validation results on 6 Hilbert-coding metasurfaces show the deflection angles of the four single beam patterns obtained are about 21°, 13°, 12°, and 39°, with energy values of 7.75 dB, 7.3 dB, 7.2 dB, and 7.7 dB, respectively, and the deflection angles of the dual-beam patterns are 28.5° and 20° with energy values of 10.05 dB and 11.4 dB, respectively. The results are quite consistent with the simulation data, further confirming the feasibility of our idea. In addition, there are potential applications in Wireless Communications and Radar-imaging, like EM beam scanning and EM field energy distribution control in communication and imaging scenarios.

## 1. Introduction

Metamaterials [1,2] are artificial composites composed of sub-wavelength-sized structures arranged in three dimensions, either periodically or non-periodically. The critical physical parameters of the structures are designed to enable the remarkable physical qualities that do not exist in natural materials, such as negative magnetic permeability [3,4,5], negative refractive index [6,7,8], and zero refractive index [9]. The advancement of electromagnetic (EM) [10] metamaterials has greatly enhanced the ability to manipulate EM fields. To date, metamaterials research has been of great value to humanity in the disciplines of stealth [11,12], wave absorption [13], and anomalous refraction [14]. As a two-dimensional version of metamaterials, metasurfaces further take advantage of the abrupt change in EM waves to achieve broader regulation of the EM field, allowing for flexible and effective design and control of EM wave amplitude [15,16], phase [17,18,19,20], polarization [21,22], and propagation modes [23].

In 2014, prof. Cui and his team at Southeast University introduced the concept of digital coding metamaterials and metasurfaces [24]. With their flexible tuning properties and simple design methods, digital metamaterials have inspired a wide range of EM functions and applications [25], including vortex beam [26], beam manipulation [27], holographic imaging [28,29], and full space scattering field tuning [30]. The passive coding metasurfaces [31] are a type of digital metasurfaces that achieve different EM functions by changing the structural parameters of passive metasurface units. For example, the V-shaped structure [32] with different tension angles and orientations realizes the functions of negative refraction, abnormal reflection, and focusing of EM waves. The patch-shape structure [24,33,34,35] is encoded for amplitude, and phase by changing the size of the patch structure, enabling anomalous reflection. The C-ring structure [36,37] is encoded for amplitude, and phase by changing the arm length and the open angle, achieving deflection and beam focusing of the transmitted wave, etc. However, these passive structural metasurfaces mentioned above are still limited to achieving specific EM responses by adjusting structural parameters, that is, coding amplitude, phase, and polarization by changing structural parameters, which still requires a complex design process. Hilbert-curve, as a simple space-filling curve constructed from iteration methods, has the potential to enable a new coding manner.

In this paper, we propose Hilbert-coding metasurface for flexible and convenient EM regulation by arranging Hilbert-coding metamaterial units of different orders. We take orders 1–4 Hilbert-coding metamaterial units as research objects, employ the numbers “1”–“4” to represent the four phase reactions of the Hilbert-coding metamaterial units, and design the phase distribution of the metasurface by this coding form. To verify our idea, we developed 12 different Hilbert-coding metasurfaces, including four metasurface arrays, four chess-board metasurfaces, two orders Hilbert-coding metasurfaces, and radar cross section (RCS) pattern metasurfaces. In addition, we measured 6 samples of the Hilbert-coding metasurfaces, and the results agree well with the simulated values. This Hilbert-coding metasurface can be arbitrarily assigned different functions and properties and has a wide range of application scenarios. Most importantly, due to the combination of special curves and digitally coding metamaterials, the Hilbert-coding metasurface provides a new method to combine geometrical-mathematical and electromagnetic modulation, which has a high potential for designing new electromagnetic applications, such as wireless communication and radar image directions.

## 2. Principle and Design

The Hilbert curve is a continuous fractal space-filling curve. It is constructed by following simple iterative rules, beginning with an element shaped like a “U”, and each level of the curve is a combination of scaling, translation, and rotation operations of the previous level to form a pattern of increasing density and complexity. Precisely, the image of Hilbert-curve is the unit square, whose dimension is 2 in any definition of dimension; its graph is a compact set homeomorphic to the closed unit interval, with Hausdorff dimension 2. The Hilbert curve is constructed as a limit of piecewise linear. The length of the *n*st curve is $2^{n}-\frac{1}{2^{n}}$, i.e., the length grows exponentially with *n*, even though each curve is contained in a square with area 1. As shown in Figure 1a, the first-order to sixth-order Hilbert curves are displayed. In this research, we code the metamaterial units by order and construct the metasurface phase distributions in this method. Here, we take orders 1–4 Hilbert-coding metamaterial units as research objects. Under the normal y-polarized plane wave illumination, the simulation frequency is at 2–14 GHz. The working frequency is 10.0 GHz, the four Hilbert-coding metamaterial units have four phases, so we represent the four phase responses using numbers “1”–“4”, and design the phase distribution of the metasurface by this coding form. After experimental verification, the Hilbert-coding metasurfaces of different orders and sizes can achieve different EM functions. For instance, as shown in Figure 1, the Hilbert-coding metasurface array coding with “11223344112233441122”, two orders Hilbert-coding metasurfaces coding with “11114444111144441111”, the Hilbert-coding chess-board metasurface coding with “11223344112233441122” in the horizontal direction, and the Hilbert-coding metasurface of RCS pattern are implemented as single-beam, dual-beam, multi-beam, and RCS reduction, respectively.

In this paper, we design 12 Hilbert-coding metasurfaces with different coding patterns composed of 20 × 20 Hilbert-coding metamaterial units. Specifically, we first design units of order 1–4 Hilbert-coding metamaterials, as shown in Figure 2. As shown in Figure 2a–d, three-dimensional models of order 1–4 Hilbert-coding metamaterial units respectively. Each independent Hilbert-coding metamaterial unit is composed of distinct Hilbert geometric metal band orders, intermediate media, and metal substrates. We design metamaterial units with a period of *p*, the thickness of the Hilbert curve-shaped belts is 0.1 mm, the dielectric constant of the intermediate layer is 4, and the thickness of the four units is h1, h2, h3, and h4. The thickness of the metal ground is 0.01 mm. In particular, taking the first order Hilbert curve-shaped belts as an example, its length is a1 and width is w1. The lengths of the U-shaped structures inside the Hilbert-coding metamaterial units of order 2–4 are a2, a3, a4, and the widths are w2, w3, and w4, respectively. In order to achieve the desired phase difference (about 90°) of the four Hilbert-coding metamaterial units, we simulate in the software CST Microwave Studio. In the simulation analysis, a more perfect set of experimental results in the y-polarization direction was produced when *p* = 10 mm, a1 = 2.5 mm, w1 = 0.8 mm, h1 = 3 mm, a2 = 1.25 mm, w2 = 0.8 mm, h2 = 3.5 mm, a3 = 0.4 mm, w3 = 0.1 mm, h3= 3 mm, a4 = 0.25 mm, w4 = 0.2 mm, h4= 3 mm. The amplitude and phase of this group of parameters are shown in Figure 2e,f. It can be seen from Figure 2f that the phase difference between the metamaterial units of order 1–4 is about 90° at the frequency point of 10.0 GHz. The simulated amplitude values of the 1–4st Hilbert-coding metamaterial unit are −0.06 dB, −0.014 dB, −0.097 dB, and −0.018 dB, respectively, and the amplitude graph is shown in Figure 2e.

To verify our idea, we conduct EM simulation on all the Hilbert-coding metasurfaces of these 12 different coding patterns, and the simulation results are shown in Figure 3, Figure 4 and Figure 5. In addition, to verify the accuracy of simulation results, we compared the beam deflection angle obtained by the simulation experiment with the deflection angle obtained by theoretical calculation. Taking the single beam deflection angle realized by the array with the code of “11223344112233441122” as an example, according to the theoretical calculation formula of the beam deflection angle of the metasurface composed of N × N metamaterial units [38]:(1)$$\theta =sin^{-1}\left(\frac{\lambda }{nP}\right)$$
where λ is the wavelength of the central frequency point, *n* is the number of units within a period, and *P* is the period length of each unit.

According to Formula (1), the theoretical deflection angle of the metasurface array with coding sequence “11223344112233441122” should be 21.09°, and in the simulation results, the deflection angle is about 21°. The error between the simulation value and the theoretical value is very small. It shows that the simulation experiment has good consistency with the theory.

In Figure 3, we give the two-dimensional plane view and simulation results of the metasurface arrays with four coding patterns. As shown in Figure 3a–d, there are the metasurface array patterns with the coding sequences as “11223344112233441122”, “11122233344411122233”, “11112222333344441111”, and “11111222223333344444”. The two-dimensional electric field results of the four different coding metasurface arrays show in Figure 3e,f, and the three-dimensional far-field diagram of the metasurface arrays show in Figure 3i–l. The simulation results demonstrate that the above metasurface arrays can realize the single-beam EM function. At 10.0 GHz, the metasurface arrays coding with “11223344112233441122”, “11122233344411122233”, “11112222333344441111”, and “11111222223333344444” have single beam deflection angles of 21°, 14°, 10°, and 8°, respectively, and have maximum beam energy values of 11.5 dB, 12.2 dB, 11.3 dB, and 11.98 dB, respectively.

As shown in Figure 4a–d, the two-dimensional plane view of Hilbert-coding chess-board metasurfaces with four coding patterns. The horizontal direction of the four chess-board metasurfaces is coded as “11223344112233441122”, “11122233344411122233”, “11112222333344441111”, and “11111222223333344444”. The two-dimensional electric field results and far-field diagrams obtained by the simulation of four coding chess-board metasurfaces are shown in Figure 4e,f,i–l. The simulation results indicate that the Hilbert-coding chess-board metasurfaces allow for multi-beams. The simulation results show that the beam deflection angle is 21° and the beam energy is about 7.75 dB for the Hilbert-coding chess-board metasurface with the horizontal direction coding as “11223344112233441122”; the Hilbert-coding chess-board metasurface with horizontal direction coding as “11122233344411122233” can achieve deflection angle at 13° and the beam energy is about 7.3 dB; the beam deflection angle is 12° and the beam energy is about 7.2 dB of the Hilbert-coding chess-board metasurface with the horizontal direction coding as “11112222333344441111”; the beam deflection angle is 39° and the beam energy is about 7.7 dB for the Hilbert-coding chess-board metasurface with the horizontal direction coding as “11111222223333344444”. 

In particular, due to the flexibility of our proposed Hilbert-coding metasurface, we also design the metasurface arrays with two orders of Hilbert-coding metamaterial units, as shown in Figure 5a–c. The metasurface of the RCS pattern is shown in Figure 5d. The metasurface arrays are encoded as “11133311133311133311”, “11111333331111133333”, and “11114444111144441111”; RCS pattern coding metasurface code is random coding. The two-dimensional electric field diagram obtained by the above four metasurface simulations is shown in Figure 5e,f, and the realized far-field diagram is shown in Figure 5i–l. The near-field results and far-field results in Figure 5e–l demonstrate that the metasurface array coding as “11133311133311133311” and “11111333331111133333” can realize dual-beam concerning normal symmetry, and the dual-beam deflection angle is 28.5° and 15°, respectively. The maximum energy values that can be realized are about 10.05 dB and 10.61 dB. The metasurface array coding “11114444111144441111” can also attain dual-beam, the beam deflection angle is 20°, and the maximum beam energy is about 11.4 dB.

We fabricate the metasurface arrays with coding “11133311133311133311”, “11114444111144441111”, “11223344112233441122”, “11122233344411122233”, “11112222333344441111”, and “11111222223333344444”, then measure the far-field results in the standard microwave chamber. It is worth mentioning that because the thickness of the middle dielectric layer of the independent units is not the same, we made the metasurface into a strip version and arranged them together during the measurement. Two sample diagram modes are shown in Figure 6a,b. Figure 6a shows the metasurface array coding as “11112222333344441111” and Figure 6b shows the sample the metasurface array coding as “111111222223333344444” in an experiment demonstration, as shown in diagram Figure 6c. The metasurface samples and a feed source are fixed on a rotatable table. Two horn antennas are the feed source and receiver, respectively. The far-field data are measured on a two-dimensional plane while the rotating table is rotated. The feed horn antenna is 1.5 m away from the metasurface, whereas the reception horn is around 10 m away from the turntable.

In Figure 7a–f, we compare the measured results and simulation results of the above six Hilbert-coding metasurface arrays, where the blue curve is the simulation result and the red curve is the measured result. The far-field results show that two orders Hilbert-coding metasurfaces achieve dual-beam, and the Hilbert-coding arrays attain single-beam, which is consistent with the simulation results. The comparison results, which include the deflection angle and the energy value of the beam, reveal that the measured results correspond well with the simulated results. The dual-beam patterns deflection angles of the two order Hilbert-coding metasurfaces are about 28.5° and 20° with energy values of 10.05 dB and 11.4 dB, respectively. The single beam patterns deflection angles of the Hilbert-coding metasurfaces array are about 21°, 13°, 12°, and 39° with energy values of 7.75 dB, 7.3 dB, 7.2 dB, and 7.7 dB, respectively. It is worth mentioning that the modest difference between measured and simulated results could be due to the following factors: (1) The slight angle deviation is due to the manual placement of horn antennas and metasurface samples; (2) the measurement system’s limited metasurface size and emitting power, which may cause interfering noise.

## 3. Conclusions

To summarize, we propose Hilbert-coding metasurfaces for flexible and convenient EM regulation. As a new regulation method, Hilbert-coding metasurface is designed to achieve various scattering field manipulation, such as dual-beam and four-beam field, by arranging different orders of Hilbert metamaterial units. Validation results on 6 Hilbert-coding metasurfaces show that the experimentally obtained angles and energy match well with the simulated results. The deflection angles of the four single beam patterns obtained are about 21°, 13°, 12°, and 39° with energy values of 7.75 dB, 7.3 dB, 7.2 dB, and 7.7 dB, respectively, and the deflection angles of the dual-beam patterns are 28.5° and 20° with energy values of 10.05 dB and 11.4 dB, respectively. We envision that, due to combining the Hilbert curve and digital coding metasurface, the Hilbert-coding metasurface has potential applications in Wireless Communications and Radar-imaging. We envision that, due to combining the Hilbert curve and digital coding metasurface, the Hilbert-coding metasurface has potential applications in Wireless Communications and Radar-imaging.

## Figures and Tables

**Figure 1 materials-15-05913-f001:**
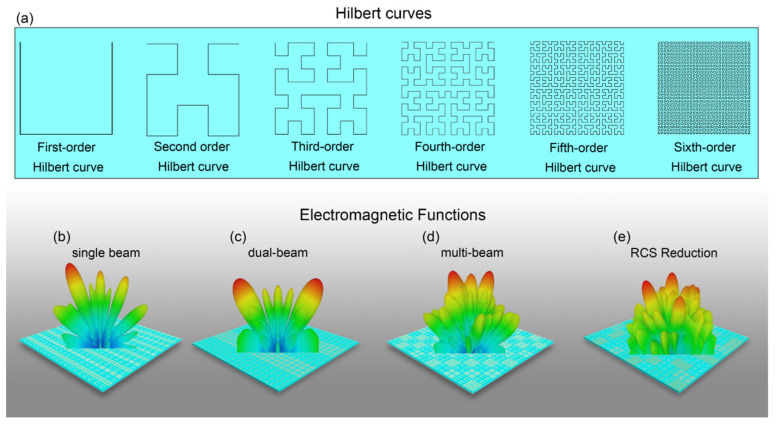
(**a**) Hilbert-curves. Including 1–6st Hilbert-curve. (**b**–**e**) Four examples of Hilbert-coding metasurfaces achieving EM functions. From left to right are the Hilbert-coding metasurface array coding with “11223344112233441122”, two orders Hilbert-coding metasurfaces coding with “11114444111144441111”, the Hilbert-coding chess-board metasurface coding with “11223344112233441122” in the horizontal direction, and the Hilbert-coding metasurface of RCS pattern, the functions achieved in order are single-beam, dual-beam, multi-beam, and RCS Reduction.

**Figure 2 materials-15-05913-f002:**
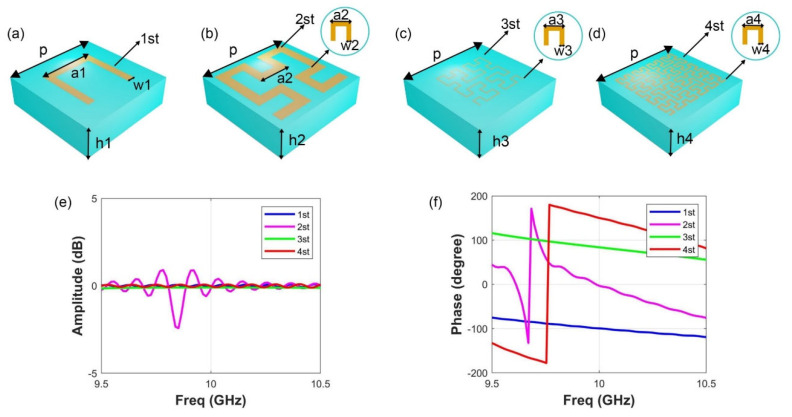
(**a**–**d**) The design diagram. Including the 3D model of order 1–4 Hilbert-coding metamaterial units; (**e**,**f**) The simulation results of order 1–4 Hilbert-coding metamaterial units in the software CST Microwave Studio; (**e**) The amplitude result, and (**f**) The phase result.

**Figure 3 materials-15-05913-f003:**
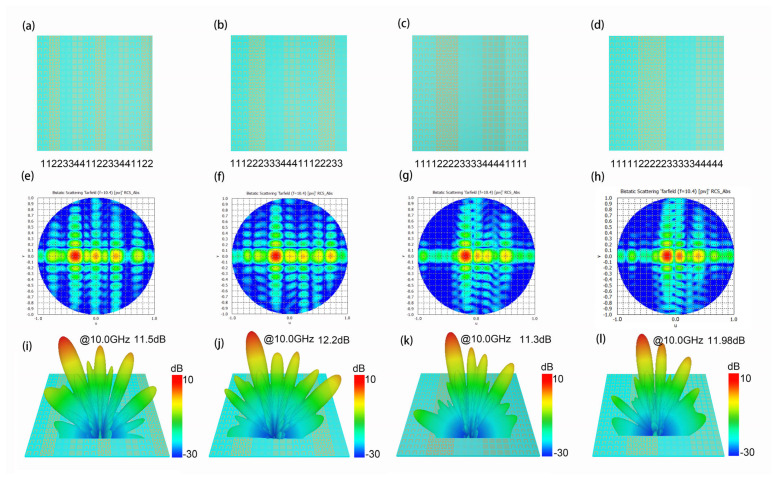
(**a**–**d**) Four different coding patterns for metasurface arrays; (**e**–**h**) The two-dimensional electric field results corresponding to the four different coding metasurface arrays; (**i**–**l**) Far-field diagrams of the metasurface arrays.

**Figure 4 materials-15-05913-f004:**
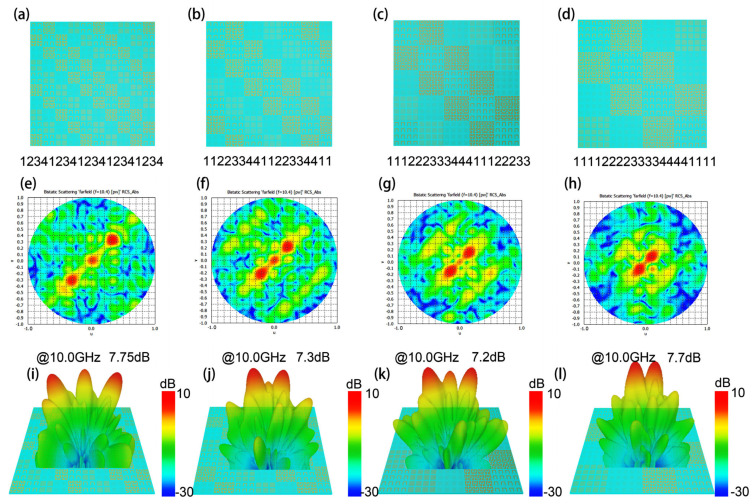
(**a**–**d**) The two-dimensional plane view of Hilbert-coding chess-board metasurfaces with four coding patterns; (**e**–**h**) The two-dimensional electric field results corresponding to the four different coding chess-board metasurfaces; (**i**–**l**) The far-field diagrams of the chess-board metasurface.

**Figure 5 materials-15-05913-f005:**
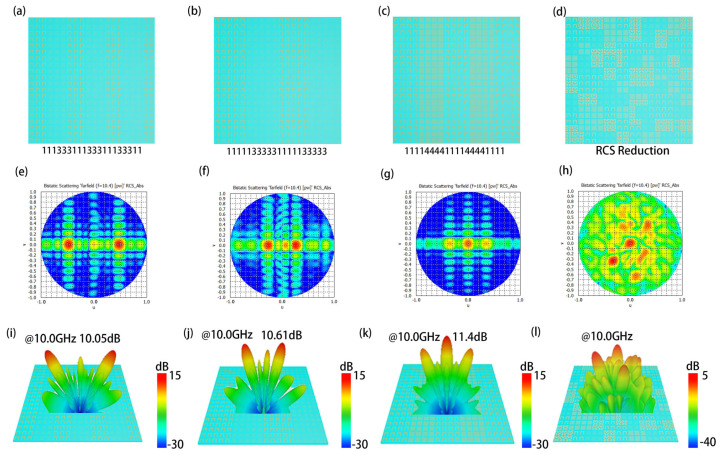
(**a**–**d**) Two orders Hilbert-coding metasurface and RCS pattern; (**e**–**h**) The two-dimensional electric field diagram obtained by the above four metasurface simulations; (**i**–**l**) Far-field diagram of the above four metasurfaces.

**Figure 6 materials-15-05913-f006:**
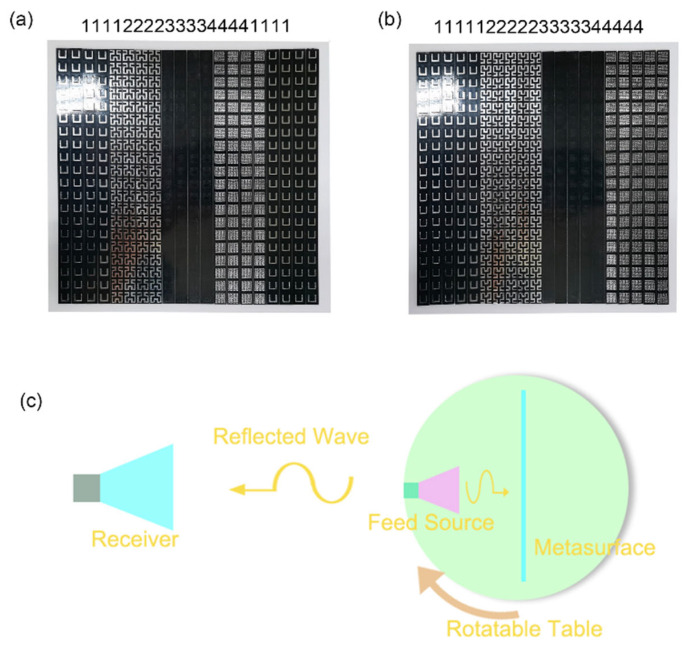
(**a**,**b**) Two sample diagram modes. (**c**) The diagram of the experiment.

**Figure 7 materials-15-05913-f007:**
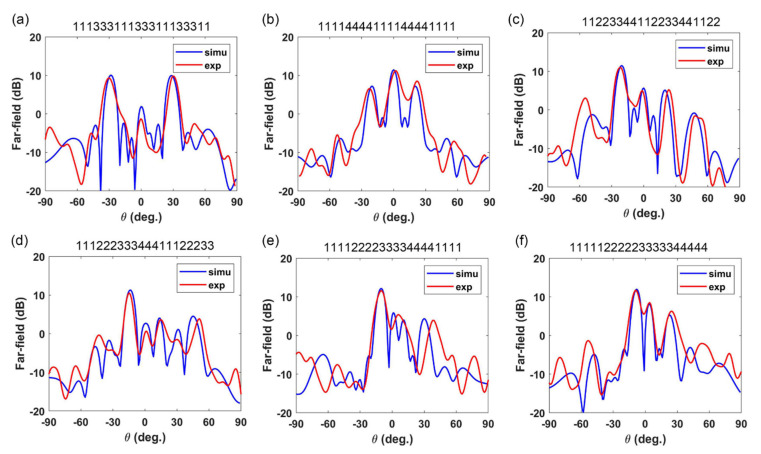
(**a**–**f**) The measured results and simulation results of the Hilbert-coding metasurface arrays. The coding sequences are “111133311133311133311”, “111114444111144441111”, “11223344112233441122”, “11122233344411122233”, “11112222333344441111” and “11111222223333344444”, respectively.

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
