# Peer review of "Hilbert-Coding Metasurface for Diverse Electromagnetic Controls"

_materials, 2022, doi:10.3390/ma15175913_

Round 1

Reviewer 1 Report

In this manuscript, the authors simulated and experimentally validated the scattering profiles of the metasurface patterns using Hilbert-coding. Hilbert-coded patterns feature reduced degrees of freedom in geometry, simplifying the structure optimization process. By carefully selecting the design parameters, desired reflection phase can be achieved at target RF frequencies. Arranging the patterns with different reflection phases periodically yielding a certain deflection angle which depends on the period, wavelength, and repeat times. The measured results matched well with the simulated results, proving the effectiveness in design prediction. To further improve the manuscript, the authors may want to improve clarity and elaborate details. Some suggestions can be found below

1)    It is better to add a figure to illustrate Hilbert curves of H1 to H4 right after the introduction of Hilbert curves in Section 2. Although Fig. 2 shows the 3D model of the device, it is one page below the part where Hilbert curves are introduced. Besides, the higher order Hilbert curves are not clearly depicted in Fig. 2.

2)    In Line 62, the description of “Under the illumination of the plane wave” is vague. Does the plane wave illuminate normally on the metamaterial device? At what frequency?

3)    In Line 89, h2=3.5 mm, while the rest designs all feature h=3 mm. How do the authors fabricate the devices with different metal height on one PCB?

4)    In Line 92, the description of “amplitude of the metamaterial unit at this frequency point is large enough” is unclear. How large is large enough?

5)    In Line 133, the beam energy is “.3 dB” whereas other configurations have beam energy around 7 dB. Is there a typo?

6)    Please indicate which patterns do Figure 6 (a) and (b) correspond to.

7)    It is better to add an experiment setup figure while describing the measurement system.

8)    The language and writing style need to be polished. For example, in Line 25, citation [3-5] follows “such as”. I assume they should follow “negative magnetic permeability”. In Line 30, “achieve boarder and sharper tailor”, I assume you want to express “tail”. “Tailor” is someone who sews cloth. By the way, “broader” and “sharper” are contradictory to be used together. In Line 46, it is confusing to read the sentence “reflection phases of Hilbert metamaterial units of different orders of the Hilbert metamaterial units”. Some acronyms like “RCS” appear without its full name spelled out.

Reviewer 2 Report

In this paper it is found, that the reflection phases of Hilbert metamaterial units are distinct and propose to encode the Hilbert coding metasurface by adjusting the order of Hilbert coding metamaterial units. I think, the paper is interesting. I would propose some changes as follows:

1.  Authors should justify usage of Hilbert coding metasurfaces.

2. Authors should stress novelty of their work.

3. It would be interesting to calculate absorption of the structures under consideration.

4. It would be desirable to enhance the Conclusions sectio  by adding more conclusions based on the obtai ed results.

2. 

Reviewer 3 Report

MDPI Materials

Materials-1821040

Hilbert-coding metasurface for diverse electromagnetic controls

1.      The application of this proposed paper in abstract section should be cleared more.

2.      Without define the RCS, authors used the abbreviation many times in the manuscript.

3.      Conclusion part is very confused. It must be revised.

4.      If the work is containing experimental results, authors should provide the experimental setup diagram or share the photograph as evident.

5.      Figs. 6 c-h are not clear. Kindly split up the plots for clear visible.

6.      Authors portrays the Fig. 2 a-d as designed diagram.

7.      Physical interpretation of Fig. 2(e) and 2 (f) are inadequate.

8.      kindly enlarge the fig 2(e) for better resolution of all color lines.

9.      Authors must provide comparison table using previously published results for proving the novelty of work.

10.  What is the importance of deflection angle in the work? And how it is related to RCS.

11.  Needed more explanation for Figs 6 (a- h) which is the main focusing in this work.

12.  In both abstract and conclusions, authors have not mentioned about deflection angle. Revise the both sections accordingly.

Major Revision required before acceptance.

Round 2

Reviewer 1 Report

My comments have been addressed.

Reviewer 3 Report

Revised manuscript is suitable for publication